# Curcumin-Loaded Self-Assembly Constructed by Octenylsuccinate Fish (*Cyprinus carpio* L.) Scale Gelatin: Preparation and Characterization

**DOI:** 10.3390/foods11182911

**Published:** 2022-09-19

**Authors:** Xiaoyan Yu, Haoxin Li, Aida Wan Mustapha Wan, Tingyuan Ren, Zunguo Lei, Jia Liu

**Affiliations:** 1School of Liquor & Food Engineering, Guizhou University, Guiyang 550025, China; 2The Key Laboratory of Environmental Pollution Monitoring and Disease Control, Ministry of Education, School of Public Health, Guizhou Medical University, Guiyang 550025, China; 3Department of Food Sciences, Faculty of Science and Technology, Universiti Kebangsaan Malaysia (UKM), Bangi 43600, Malaysia; 4Institute of Food Processing Technology, Guizhou Academy of Agricultural Sciences, Guiyang 550006, China

**Keywords:** curcumin, fish scale, hydrophobic interaction, nano-assembly, octenylsuccination

## Abstract

Curcumin loaded octenylsuccinate fish scale gelatin (OFSG) was prepared in this study, to explore the potential of FSG for delivering hydrophobic nutrients. The effects of molecule weight (*M*_w_, 22,677–369 g/mol) and degree of substitution (*DS*, 0–0.116) on the curcumin loading efficiency (CLE, μg/mL) of OFSG (6.98–26.85 mg/mL) were evaluated. The expose of interior hydrophobic groups in FSG and increased intermolecular hydrophobic area contributed to the loading of curcumin in two phases, respectively. The interaction between OFSG and curcumin showed a decreased absorption in FTIR and an increased crystallinity in XRD. The loading of curcumin into OFSG caused a significant decrease of the particle size (from 350–12,070 to 139–214 nm), PDI (from 0.584–0.659 to 0.248–0.347) and *ζ*-potential (−12.2 or −11.4 to −21.0 or −20.3). OFSG showed a significantly higher stability and lower release of curcumin than FSG at the end of the simulated gastrointestinal digestion. Thus, OFSG showed great potential in the construction of a carrier for hydrophobic nutrients.

## 1. Introduction

Fish scales (FS) are a natural substance of extremely solid quality and act as the armor or coat for fishes living in lakes, rivers and oceans. The biomaterial in FS is mainly composed of type I collagen, which endows its application potential in food, medicine, chemical engineering and other industries [1,2,3,4]. However, the commercial utilization of FS still lacks the participation of enterprise due to its investment with high cost during the extraction and purification of biomaterials. Unfortunately, FS are often discarded in different markets and even fish canning, filleting, salting and smoking [5]. According to the statistics, millions of tons of FS are discarded in the world per year and cause serious environmental pollution [2]. Thus, technology with lower cost by converting FS into a commercial product is highly demanded and can significantly minimize the problem of environmental pollution.

Gelatin is derived from heat-denatured collagen and widely used in the food industry [2]. Normally, the application of gel formed with fish gelatin (FG) was limited due to its lower strength, gelling and melting temperature. Thus, the physicochemical properties of FG from skin or scale could be enhanced by modification with acyl gellan [6], sodium alginate [7] and *κ*-carrageenan [8]. Fish scale gelatin (FSG) can be easily obtained by the aid of aqueous extraction with hot water. Therefore, the broad market and low-cost acquisition causing the application of FSG in food processing deserve extensive exploration. A gel food could be acquired by the cooling of FSG close to 0 °C [9] and further structurally enhanced by mixing with transglutaminase [2,10], pectin [10,11], alginate [12] and sodium carboxyl methyl cellulose [12]. An environmentally friendly edible film could be prepared by mixing FSG with plasticizer [13] and chitosan [14]. The structure of FSG could be tailored to stabilize fish oil-loaded emulsion [4] and promote foaming properties of egg white protein [15]. The hydrolysis of FSG with an alcalase enzyme could produce hydrolysates with emulsion and antioxidant properties [16] or mixing with tannic acid to form hybrid nanoparticles [11].

These explored applications of FSG did not involve structured derivatives with exogenous groups. The grafting of diverse groups into fish gelatin (FG) would endow itself with functional characteristics and advance the new frontier for its application. Thus, continuous study followed this to explore the structural modification of FG [1,3,15,17,18,19,20]. Fish skin gelatin was modified by oxidized caffeic acid, ferulic acid and tannic acid [21] and crosslinked with glutaraldehyde [17] and *γ*-polyglutamic acid [18] for improving its antioxidative properties, gel strength and melting point. Glycosylated FSG with Arabic gum [3] and crosslinked FSG with pectin [2] were developed to improve their gelation points and emulsion properties. Gallic acid-modified fish gelatin was applied to improve the mechanical performance of the film [19]. Recently, octenyl succinic anhydride-modified fish skin gelatin was synthesized and applied to form a stable emulsion for the delivery of lipophilic bioactive substances [20]. An increase in the chain length of the alkenyl derivatives could impart greater thermal stability [22], surface activity and mechanical properties of the films [23] to proteins. Additionally, through the formation of a linear amphiphile to stabilize the interface of the oil–water system, the hydrophobic modification of hydrophilic biomacromolecules might also form a self-aggregate through the aid of hydrophobic interaction in the water system [24]. However, no literature has investigated the application of self-assembly formed by hydrophobic FSG at present. 

Thus, this study aimed to characterize curcumin-loaded self-assembly constructed by octenylsuccinate fish (*Cyprinus carpio* L.) scale gelatin (OFSG). The effects of the structural parameters (Molecular weight, *M*_w_, and degree of substitution, *DS*) were investigated on the curcumin loading efficiency of OFSG. Fourier-transform infrared spectroscopy (FTIR), X-ray diffraction (XRD), nuclear magnetic resonance (NMR), transmission electron microscopy (TEM), dynamic light scattering (DLS) and *ζ*-potential were used to characterize curcumin-loaded FSG and OFSG (CL-FSG and CL-OFSG). Finally, the curcumin release and stability of CL-FSG and CL-OFSG in simulated gastrointestinal digestion were also evaluated. This study explores the application of OFSG in forming self-assembly and its capability for loading lipophilic bioactive substances, which would be conductive in the development of FSG based on functional foods.

## 2. Materials and Methods

### 2.1. Materials 

Fresh fish (*Cyprinus carpio* L.) scales (FS) were obtained from the experimental base at Guizhou Fisheries Research Institute (Guiyang, China). FS were washed with tap water 3 times and placed in an oven (WGL-65B, Taisite Instrument Co. Ltd., Tianjin, China) at 40 °C for 20 h. The dried FS were finally stored at 15 ± 0.5 °C in a Ziplock plastic bag and placed in a dry environment until analysis. Curcumin (purity > 90%) was purchased from Merck-Schuchardt Co. (Hohenbrunn, Germany). 2-Octen-1-ylsuccinic anhydride of 210.26 g/mol was purchased from Sigma Chemical Co. (St. Louis, MO, USA). Trifluoroacetic acid (analytical grade) and acetonitrile (chromatographic grade) were purchased from Chron Chemicals Co,. Ltd. (Cheng Du, China). Pepsin (4000 U/g) and pancreatin (3000 U/g) were purchased from Solarbio Science & Technology Co., Ltd. (Beijing, China) and Yuanye Science & Technology Co., Ltd. (Shanghai, China). All chemicals were analytical reagents without further purification.

### 2.2. FSG Extraction and OFSG Synthesis

The extraction of FSG was conducted as the former method [12]. The dried FS were ground (S4-M71, Jouyang Co., Ltd., Shandong, China) and then immersed in a flask with distilled water (FS:water = 1 g:15 mL) with a magnetic stirrer (90 ± 1 °C, 1 h). The mixture was cooled and centrifuged (L550, Xiangyi Science & Technology Co., Ltd., Hunan, China) at 4000× *g* for 5 min. The supernatant fluid was dialyzed with distilled water for 24 h in a dialysis bag (MD44-14, Union Carbide Co., Seadrift, TX, USA) with a size exclusion of 14,000 g/mol for globular molecules. After dialysis, FSG was frozen at -18 ± 1 °C for 2 h, followed by lyophilization (10YG/A, SCIENTZ Co., Ltd., Ningbo, China) for 24 h. 

The synthesis of octenylsuccinate-based FSG was conducted by the method described before [24]. FSG solution (2%, *w*/*v*) was prepared by dissolving dried FSG in distilled water at 70 °C for 5 min. Then, 1%, 2%, 4%, 5%, 7% and 10% (*w*/*v*) OSA were added in FSG solution at 15 ± 1 °C to obtain OFSGs with different *DS* and *M*_w_. The pH of the mixture was adjusted to 8.0–10.0 by the continuous addition of 3% (*w*/*v*) NaOH. At the end of the reaction, 2 mol/L HCl was added to the solution until the pH reached about 6.5. The solution was continuously stirred for 12 h at 15 ± 1 °C. The resulting solution was centrifuged at 8000× *g* for 10 min, and the supernatant was dialyzed for 24 h against tap water to remove the residual OSA. The dialyzed samples were freeze-dried and stored at 4 °C for further analysis of the *DS* and *M*_w_.

### 2.3. Determination of DS and Mw of OFSGs

The degree of substitution was determined according to Qiu, Bai and Shi (2012) [25], with some modifications. Dried OFSG (0.050 ± 0.005 g) was immersed in 10 mL of 4 M NaOH and stirred at 15 ± 1 °C overnight. The sample for the determination of *DS* was composed with a 2 mL alkali-treated solution, 10 mL HCl (1 M) and 13 mL acetonitrile. The samples were vortexed for 2 min and centrifuged at 5600× *g* for 10 min. The supernatant (20 μL) was filtered through a 0.45-μm filter (Nylon 66, Shanghai, China) and injected into an Agilent 1260 Infinity HPLC system (Agilent Technologies, Santa Clara, CA, USA) with the UV spectra at 200 nm. A reverse-phase Thermo BDS C18 column (250 mm × 4.6 mm i.d.) from Thermo Fisher Scientific Inc. (Waltham, MA, USA) with a 5-μm particle diameter and a mixture of acetonitrile and ultrapure water (55:45, *v*/*v*) containing 0.1% trifluoroacetic acid was used in the analysis system. In order to exclude the free OSA attached to the sample, OFSG was firstly immersed into 5 mL methanol and continuously stirred for 1 h. The supernatant was obtained by centrifuging for 10 min at 3000× *g* and followed with the determination of the free OSA concentration by HPLC. The OSA concentrations of OFSG with and without the alkali treatment were recorded as C_1_ and C_2_, respectively. The *DS* of OFSG was calculated using following equations:
OS%(total) = (C_1_ × 125/W) × 100%(1)
OS%(free) = (C_2_ × 10/W) × 100%(2)
OS% = OS%(total) − OS%(free)(3)
*DS* = (162 × OS%)/(100 × 210 – 209 × OS%)(4)
where C_1_ and C_2_ are the OSA concentrations in OFSG calculated from the standard curve (mg/mL) with or without the alkali treatment, 125 and 10 are the dilution factors and W is the dry weight of the sample (mg). The determination of FSG and OFSG *M*_w_ was carried on a gel column (TSKgel GMPWXL, Tosoh Bioscience GmbH, Griesheim, Germany) and a high-performance size exclusion chromatography system [26]. Different *M*_w_ values (1 × 10^3^, 4 × 10^4^, 10 × 10^4^, 50 × 10^4^ and 200 × 10^4^ g/mol) of polyethylene glycol were used as the *M*_w_ standard reference. 

### 2.4. Preparation of CL-FSG and CL-OFSG

The freeze-dried FSG and OFSG (40 mg) were firstly dissolved in 20 mL distilled water by heating to a boiling state for 1 min [27]. Then, the solution was cooled to 15 ± 1 °C when the sample was completely dissolved, and the volume was increased to 20 mL with distilled water. Curcumin powder (1 mg) was added into a 20 mL FSG or OFSG solution. Then, the solution was homogenized at 12,000 rpm for 1 min using a homogenizer (IKA T18 basic, IKA-Werke GmbH & Co., Staufen, Germany). For the purpose of sufficient contact between curcumin and FSG or OFSG, the homogenized suspension was continuously stirred at 15 ± 1 °C in bath water with 300 rpm agitation for 24 h. Finally, the solution was centrifuged at 5600× *g* for 10 min, and the supernatant was collected for further analysis.

### 2.5. Determination of Curcumin Concentration in CL-FSG or CL-OFSG

The curcumin concentration of the CL-FSG or CL-OFSG solution was determined by HPLC [24]. Curcumin in hydrophobic cavity was extracted by mixing CL-FSG or CL-OFSG solution with methanol (1:4, *v*/*v*). Mixtures were vortexed for 2 min, centrifuged at 5600× *g* for 10 min and filtered through a 0.45-μm filter (Nylon 66, Shanghai, China). The separation of curcumin was conducted with a reverse-phase Thermo BDS C18 column (250 mm × 4.6 mm i.d.) from Thermo Fisher Scientific Inc. (Waltham, MA, USA) with a 5-μm particle diameter at 25 ± 1 °C. An Agilent 1260 Infinity HPLC system (Agilent Technologies, Santa Clara, CA, USA) with a DAD detector at 420 nm was used for the determination of the curcumin concentration. Formic acid (solvent A, 0.2%, *v*/*v*) and acetonitrile (solvent B) were mixed and supplied as the mobile phase at a flow rate of 0.7 mL/min. The gradient program for the separation of curcumin was performed as: 0 min, A:B, 65%:35% (*v*/*v*); 0–10 min, B increased from 35% to 65%; 10–15 min, B increased from 65% to 70% and 15–20 min, B increased from 70% to 75%. The curcumin loading efficiency (CLE, μg/mL) was defined as the mass of curcumin (μg) in 1 mL FSG or OFSG solution and quantified by an external calibration curve.

### 2.6. Characterization of FSG and OFSG with or without the Loading of Curcumin

The sample powders of FSG, OFSG_1_ (*DS* = 0.026) and OFSG_2_ (*DS* = 0.107) were obtained by drying with a vacuum freeze-dryer (10YG/A, SCIENTZ Co., Ltd., Ningbo, China) at −60 °C for 12 h and used for the subsequent characteristic analysis.

The curcumin loading capacity (CLC, μg/mg) of FSG and OFSG_1,2_ (in 1 mL of CL-FSG or CL-OFSG solution) were determined as described and calculated by using the following equation:CLC (μg/mg) = mass of curcumin (μg)/mass of FSG or OFSG (mg) (5)

The Fourier-transform IR (FTIR) spectra (from 500 to 4000 cm^−1^) of FSG and OFSG_1,2_ with or without the loading of curcumin was acquired with the KBr disk method and recorded by using a FTIR 2000 spectrophotometer (Perkin-Elmer, Norwalk, CT, USA) with a deuterated triglycine sulfate detector [18].

The X-ray diffraction (XRD) patterns (from 6 to 30°) of FSG and OFSG_1,2_ with or without the loading of curcumin were analyzed with an X-ray polycrystal line powder diffractometer (D8 ADVANCE, Brugger, Germany) at a scanning speed of 2°/min with a voltage of 40 KV [28].

The ^1^H Nuclear Magnetic Resonance (NMR) spectra (from 0 to 6 ppm) of FSG and OFSG_1,2_ with or without the loading of curcumin was recorded in a 600 MHz Bruker AVANCE-ⅢNMR spectrometer [29,30]. Samples (0.5 mg) were weighed and dissolved in 1.0 mL D_2_O. Then, the solution was centrifuged at 5000× *g* for 10 min to remove the insoluble material. The supernatant liquid was transferred into 5-mm tubes for the NMR analysis, and tetramethylsilane (TMS) was used as the standard at 0 ppm.

### 2.7. Dynamic Light Scattering (DLS) and ζ-Potential

Zetasizer Nano ZS90 (Malvern Instruments, Worcestershire, UK) equipped with a He-Ne laser (633 nm) and a scattering angle of 165° was used for the determination of the particle size, polydispersity index (PDI) and *ζ*-potential of FSG and OFSG with or without the loading of curcumin [24]. The intensity-weighted harmonic mean particle diameter (Z-Average) and polydispersity index (PDI) were determined based on DLS (the refractive index for the mobile phase was 1.33). The *ζ*-potential was determined based on the particle surface charge. The results of the particle size and *ζ*-potential were determined at 25 ± 0.5 °C and expressed as nm and mV.

### 2.8. Transmission Electron Microscopy (TEM)

The morphology of FSG and OFSG with and without the loading of curcumin was characterized by TEM (JEM 1200EX, JEOL, Tokyo, Japan) [24]). The freshly prepared sample was diluted 10 times with deionized water. Then, one drop of a diluted sample was placed on a freshly glow-discharged carbon film on a 400-mesh copper grid and stained with 1% uranyl acetate. The observations were imaged with a Tungsten filament lamp at 100 kV and 15,000× magnification.

### 2.9. Curcumin Stability and Release of FSG and OFSG in Simulated Gastrointestinal Digestion 

Simulated gastric fluid (SGF) was composed of 2.0 g NaCl, 7.0 mL HCl (37%), 900 mL double-distilled water, and 100 mL enzyme solution contained 3.2 g pepsin [31]. Simulated intestinal fluid (SIF) was mixed by 650 mL KH_2_PO_4_ (6.8 g), 190 mL 0.2 N NaOH and 160 enzyme solution and contained 10.0 g pancreatin [31]. The final pH values of the SGF and SIF were adjusted to 3.0 ± 0.2 and 7.2 ± 0.2, respectively. The sample was firstly subjected to the simulated gastro digestion in SGF and then followed with simulated intestinal digestion in SIF.

(1) For the determination of curcumin stability, 4 mL CL-FSG or CL-OFSG was firstly mixed with 16 mL SGF or SIF in a glass breaker (covered with a plastic wrap) and then incubated in a 37 ± 1 °C water bath with 120 rpm agitation. At predetermined time points (0, 10, 20, 30, 40, 60 and 90 min for SGF and 0, 10, 30, 60, 120, 240 and 360 min for SIF), 0.1 mL of the mixed solution (0.1 mL SGF or SIF was added back into the mixed solution) was sampled to determine the curcumin content by HPLC. 

(2) For the determination of curcumin release, 4 mL CL-FSG or CL-OFSG was firstly mixed with 16 mL SGF or SIF and filled a dialysis bag (a molecular weight cut-off of 1000 Da). Finally, the dialysis bag immersed in a glass breaker (covered with a plastic wrap) with 50 mL of Tween 80-incorporated digestive fluid (10%, *v*/*v*) was incubated in 37 ± 1 °C bath water with continuous stirring at 120 rpm. At predetermined time points (0, 10, 20, 30, 40, 60 and 90 min for SGF and 0, 10, 30, 60, 120, 240 and 360 min for SIF), 1 mL solution outside the dialysis bag (1 mL releasing media was added back) was withdrawn to determine the curcumin by HPLC. The curcumin concentration (%) = (CC_p_/CC_0_) × 100%(6)where CC_p_ and CC_0_ were the curcumin content determined at predetermined time points (μg/mL and 0 min (μg/mL), respectively.

### 2.10. Statistical Analysis

The results were expressed as the mean with a standard deviation from at least three measurements. SPSS 20.0 was used to analyze the data. ANOVA was performed to determine the least significance at *p* < 0.05 by Tukey’s HSD test [32,33].

## 3. Results and Discussion

### 3.1. Effect of Molecular Weight (M_w_) and Degree of Substitution (DS) on the CLE (Curcumin Loading Efficiency, μg/mL) of OFSG

The extracted FSG (total amino acids, 85.2%) consisted of a typical molecular weight (*M*_w_) distribution of gelatin with *α*-chains (*α*_1_ and *α*_2_, 130–100 kDa) and *β*-chains (~250 kDa), which were reported in our previous study [12]. This was in accordance with the reported structure of gelatin from snakehead (*Channa argus*) scales [34] and grass carp fish (*Ctenopharyngodon idella*) scales [35]. The OFSG was obtained through the esterification reaction between FSG and OSA. During the esterification reaction, the acid environment caused the significant degradation of the *M*_w_ of FSG from 2.3 × 10^4^ (*DS* = 0) to 419 (*DS* = 0.085) g/mol (Figure 1). However, the *M*_w_ of OFSG kept at a stable structural level from 419 to 369 g/mol with the further addition of OSA. Gelatin would easily go through a degradation process under a strong acid or alkaline environment [36]. However, the grafting of a hydrophobic group to FSG probably caused the formation of random coiling assembly, which helped it resist the environmental action to the gelatin molecule compared with the unfold one. Different from the variation of *M*_w_ of the OFSG, the increase of the addition of OSA caused the increase of *DS* from 0.026 to 0.116. The reaction efficiency of the grafting of the hydrophobic group to FSG went through a tendency to, first, increase (from 0.026/mL to 0.029/mL) and then decrease (from 0.029/mL to 0.017/mL). A further increase of OSA addition up to 10 mL caused a significant decrease of *DS* to 0.011. This observation (the decrease of *DS*) may prove the esterification reaction is a reversible reaction, implying that an esterification reaction can go through a reverse direction with excessive substrates (OSA). The powder yield (%, mg/100 mg) of OFSG with different *DS* (0.026, 0.058, 0.085, 0.107, 0.116 and 0.011) was 57.3%, 50.0%, 41.7%, 37.3%, 37.2% and 36.8%, respectively.

In the meantime, the CLE (μg/mL) of FSG and OFSG was detected by the aid of HPLC. As shown in Figure 1A, it is obvious that FSG can load curcumin into its neutral hydrophobic area without the grafting of a hydrophobic group. Although the aqueous extract of FSG from FS is mainly constructed by hydrophilic backbone structures, many hydrophobic amino acids (such as phenylalanine, proline and alanine) still exist in FSG [29]. The hydrophobic interaction between curcumin and the hydrophobic amino acid residue of the FSG caused a result of CLE (μg/mL) with 6.98 μg/mL when the interior structure of FSG was unfolded after the solubilization of gelatin in distilled water [37]. While CLE (μg/mL) of OFSG was significantly higher than that of the original gelatin, a significant increase of CLE up to 24.67 ± 1.25 μg/mL was observed with the increase of *DS* from 0 to 0.085. This could be attributed to the increased attachment of curcumin molecules to the OSA portion of OFSG [38]. However, a further increase of *DS* from 0.085 to 0.116 and a slight variation of CLE (μg/mL) was observed in a range of 24.67 ± 1.25 to 26.85 ± 1.46 μg/mL. In the initial stage, the increase of *DS* of OFSG accompanied the decrease of *M*_w_, which also increased the chance to form a hydrophobic area (free movement of the gelatin molecule was increased). However, the *M*_w_ of OFSG did not show a sharp decrease with a further increase of the *DS* from 0.085 to 0.116, which means that the molecular mobility of OFSG is fixed, and the hydrophobic area cannot increase with the limited molecular mobility (even the *DS* of OFSG increased).

The CLE (μg/mL) of FSG and OFSG with different *M*_w_ was summarized in Figure 1B. It is worth mentioning that the variations of *M*_w_ during chemical modification, including the octenyl succinic anhydride modification of fish gelatin [20] and oxidized linoleic acid modification of fish skin gelatin, are easily ignored, and all the experimental phenomena are ascribed to the variation of *DS* of modified gelatin [39]. Actually, a significant increase of CLE (μg/mL) from 6.98 ± 0.52 to 24.67 ± 1.25 μg/mL of OFSG was ascribed to two structural factors, as shown in Figure 1B: the decrease of *M*_w_ (phase 1) and the increase of *DS* (phase 2). In order to make sure which structural factor mainly affected the CLE (μg/mL) of OFSG, a series of FSG with decreased *M*_w_ (10,061, 3688 and 688) was obtained by acid treatment. It was obvious that phase 2 was shown in the comparison of CLE (μg/mL) between OFSG and FSG. The increase of CLE (μg/mL) of OFSG mainly depended on the decrease of gelatin *M*_w_ during phase 1, since the CLE (μg/mL) of FSG and OFSG showed a similar tendency. This might be attributed to the exposure of interior hydrophobic groups in the advanced structure of FSG during the decrease of gelatin *M*_w_, which increased the possibility for interactions between the curcumin molecule and hydrophobic groups [40]. However, the increase of CLE (μg/mL) of OFSG mainly depended on the increase of *DS* during phase 2. A further decrease of gelatin *M*_w_ would cause an increase in molecular mobility, which was adverse to the formation of an intramolecular hydrophobic area [21]. Nevertheless, the hydrolyzing of gelatin molecules increased the portion of hydrophilic groups (one carboxylic acid and one NH^3+^ group release with the cleavage of each peptide bond), which also caused the decrease of CLE (μg/mL) of FSG. Thus, intermolecular hydrophobic interactions among the OFSGs become the dominant loading channel for curcumin, accompanied with the increase of the *DS* of OFSG. 

### 3.2. Characterization Analysis of FSG and OFSG with or without the Loading of Curcumin

The curcumin loading capacity (CLC, μg/mg) of FSG and OFSG_1,2_ were determined as 3.49, 8.34 and 13.43 μg/mg. Compared with the CLC (μg/mg) of OSA-modified starch (1.8 μg/mg), *β*-casein (2.8 μg/mg), hydroxypropyl-*γ*-cyclodextrin (3.4 μg/mg) and *β*-glucan (3.9 μg/mg), the OFSG showed an excellent loading capacity (8.34–13.43 μg/mg) for curcumin [24,38,41,42].

FTIR spectroscopy was generally explored to analyze the functional groups and structure of gelatin [43]. The FTIR spectra of the FSG and OFSG with different *DS* are shown in Figure 2A. A strong broad band at 3427 cm^−1^, resulting from amide A (-OH and/or-NH_2_) stretching [44], can be seen in the spectra of both the FSG and OFSG. The FTIR spectra of FSG showed characteristic amide-I, amide-Ⅱ and amide-III bands at approximately 1633, 1560 and 1244 cm^−1^ [45]. The location of amide-I was shifted from 1633 cm^−1^ to 1648 cm^−1^ after the OSA modification of FSG. This variation might indicate a change in the secondary structure of gelatin [46]. To further comprehend the change of the gelatin structure after the OSA modification, curve fitting was semi-empirically processed based on the reported results (Appendix A). Several shoulder peaks indicating the secondary structures of proteins hid in the amide-I band and could be estimated by the aid of the deconvolution of amide-I [47]. Secondary structures, including α-helix, *β*-sheet, *β*-turn and random coil, were assigned to band area I (between 1654 and 1658 cm^−^^1^); band area Ⅱ (between 1642, 1638, 1632, 1627 and 1624 cm^−^^1^); band location I (at 1688, 1680, 1672 and 1666 cm^−^^1^) and band location Ⅱ (at 1648 cm^−^^1^), respectively [48]. The results after the calculations are recorded in Table 1. There are four kinds of basic secondary structures (*α*-helix, *β*-sheet, *β*-turn and random coil) that exist in the FSG and OFSG. Among which, random coil did not show a significant difference between the FSG and OFSG, while a significant increase of *α*-helix from 13% to 28%, decrease of *β*-sheet from 52% to 31% and increase of *β*-turn from 20% to 26% were observed (Table 1). However, the grafting of OSA to the FSG molecule and hydrolyzing of FSG took place simultaneously during the chemical modification. Thus, the change of secondary structure of FSG was caused by a complex reaction system (both the decrease of *M*_w_ and increase of *DS* during the modification). The transition among four different secondary structures of peptides could affect the final formation of the nanostructures through diverse self-assembly pathways [49]. Thus, the transition of *β*-sheet to *α*-helix might also contribute to the self-assembly of OFSG.

In addition, new absorption peaks were found in OFSG_1,2_, indicating the variation of the gelatin structure. The strong peak at 1560 cm^−1^ of the OFSG was ascribed to the symmetrical stretching vibrations of -COO^-^ [50]. Nevertheless, the amide-III peak of OFSG was more obvious than FSG, which may be due to N–H deformation from amide linkages, indicating the grafting of the OSA molecule to FSG [51]. A weak band of amide B at 2958 cm^−1^ was related to the N–H vibrations and characteristic absorption peaks at 2927 and 2856 cm^−1^ corresponding to CH_2_ asymmetrical and CH_2_ symmetrical stretches [50]. Two significant peaks in the FTIR spectra (N–H vibration and CH_2_ symmetrical stretches) were weakened after the loading of curcumin into OFSG (Figure 2D). This observation might show that the hydrophobic CH_2_ chain took part in the noncovalent interaction with curcumin. The abundant absorption in the fingerprint region of the FTIR spectra from 937, 1360 and 1450 cm^−1^ were also ascribed to the loading of curcumin into OFSG. This observation was even more obvious in the comparison between the spectra of FSG and CL-FSG.

According to the report from Sha et al. [52], the triple-helix structure and the single left-hand helix chain in fish gelatin exhibited two diffraction peaks around 7° and 20°, respectively. However, the FSG only displayed one characteristic diffraction peak between 17° and 20°, as shown in Figure 2B. This observation might be due to the destruction of the triple-helix structure in FSG after the high-temperature extraction process. It is obvious that the ordered state of FSG would be formed during the process of OSA modification, which caused two sharp diffraction peaks to appear in the spectra of OFSG_1_ (9.13 and 19.9) and OFSG_2_ (9.03 and 19.63) and the increase of crystallinity from 3.59% (FSG) to 10.77% (OFSG_1_) and 12.10% (OFSG_2_). A high crystalline structure of curcumin with the characteristic seven peaks at a diffraction angle of 2*θ* (8.89°, 14.48°, 17.22°, 18.18°, 23.3°, 24.60° and 25.52°) was reported [27]. After the loading of curcumin into FSG, characteristic peaks of curcumin disappeared in the XRD spectra, and a broad peak was observed in the XRD spectra between 18° and 27°. This observation indicated that intermolecular interactions between curcumin and FSG might take place, which caused the decrease of crystallinity from 3.59% (FSG) to 0.87% (CL-FSG). A similar result was observed after the loading of curcumin into OFSG (Figure 2E). The two original peaks in the XRD spectra of OFSG decreased to only one broad peak in that of CL-OFSG, which caused a decrease in crystallinity from 10.77% (OFSG_1)_ and 12.10% (OFSG_2_) to 3.47% (OFSG_1_) and 4.16% (OFSG_2_). On the one hand, the loading of curcumin into FSG or OFSG assembly was involved in the participation of a hydrophobic amino acid and OSA group, which might enlarge the amorphous region of curcumin. The transformation of crystalline to amorphous structures could improve the solubility and bioavailability of curcumin. On the other hand, the inhibited crystallization in the XRD pattern of FSG and OFSG could largely be due to the encapsulation of curcumin caused by transformation from a crystal to an amorphous state [28].

A ^1^H NMR spectrum of the OFSG and FSG in D_2_O is shown in Figure 2C. The resonance of the terminal methyl protons of the OSA was close to 0.84 ppm. The multiple peaks around 1.14 ppm corresponded to the methylene protons on the octenyl chain of the grafted OS groups [53]. The observed peak at a chemical shift of 2.00–2.30 ppm was attributed to the acetyl groups [54]. The resonance of the protons on the CH_3_ of acetyl groups could also be readily identified as close to 2.50 ppm [29]. The protons on the C=C double bond of the OSA were close to 5.26 and 5.32 ppm, respectively [29,55]. However, these peaks did not appear in the ^1^H spectrum of FSG, indicating that OSA groups were successfully grafted onto the FSG. Compared with the NMR of FSG, less peaks and a decreased intensity were shown in CL-OFSG (Figure 2F), indicating that the interaction between curcumin and OFSG mainly participated in the newly grafted OSG groups. 

### 3.3. Self-Assembly Observation of FSG and OFSG with or without the Loading of Curcumin

The self-aggregation of FSG and OFSG with or without the loading of curcumin was further characterized using TEM and the dynamic light scattering (DLS) technique (Figure 3 and Table 1). The appearance of FSG self-assembly (Figure 3A) was obviously different from OFSG_1,2_ self-assembly (Figure 3B,C). Self-assembly formed with FSG showed a solid structural particle with nonuniform distribution and without a hollow core, while the grafting of OSA into FSG caused the formation of nanoparticles with core–shell structures (Figure 3B,C) and a narrower particle size distribution (PSD) from 1000–2000 nm to 125–275 nm (Figure 3G–I). 

The particle size, particle size distribution (PDI) and *ζ*-potential of the FSG (Table 1) were 1408.3 nm, 0.824 and −25.2 mV, respectively, which was in accordance with the appearance of the TEM results showing a large size and polygynous self-assembly, while the grafting of the OSA groups into FSG exerted a significant effect on the particle size, PDI and *ζ*-potential of the aggregate. The increase of the *DS* from 0 to 0.107 caused a significant decrease of the particle size from 1408 nm to 350 nm and PDI from 0.824 to 0.584. This variation of self-aggregate shape and homogeneity was ascribed to the grafting of the OSA groups into FSG, which caused the enhancement of hydrophobic interaction among OFSGs [56], while the decrease of *M*_w_ during the chemical modification also contributed to the decrease of the particle size and PDI. Since polymers with lower *M*_w_ possessed a lower steric hindrance and higher flexibility, facilitating access to self-assembling [27], the *ζ*-potential, which provided the information on the surface properties of self-aggregates, increased from −25.2 to −12.2 with the increase of the *DS* from 0 to 0.107. The decrease of the *ζ*-potential can probably be ascribed to two reasons: (1) the significant decrease of *M*_w_ from 22,677 to 420 g/mol, which accompanied the decrease of the negative charge density on the surface of self-aggregates, and (2) the formation of new NH^3+^ groups as a result of the hydrolysis of gelatin molecules. The TEM of the OFSG also presented self-aggregates with more homogeneity after the grafting of the OSA groups.

The TEM observation showed that the loading of curcumin into FSG and OFSG caused a subtle change of appearance of the self-assembles (Figure 3D–F), while a dynamic light scattering (DLS) analysis showed the significant decrease of the particle size and PDI and increase of the *ζ*-potential after the loading of curcumin into FSG and OFSG (Table 2). This result proved that a stronger molecular interaction between curcumin and gelatin took place. Since the insoluble characteristic of curcumin, the molecular interaction between curcumin and gelatin, was mainly dominated by a hydrophobic interaction, the hydrophobic amino acids in the FSG and OSG groups in OFSG constructed a hydrophobic area and hydrophobic core for the loading of curcumin [57]. When curcumin was loaded into the hydrophobic area or hydrophobic core, the strong hydrophobic interaction attracted by hydrophobic amino acids and the OSG groups resulted in the aggregation of the FSG and OFSG (as the decrease of the particles and PDI implied). However, the outer information revealed by *ζ*-potential showed the opposite tendency. The net negative charge of FSG decreased from 25.2 to −18.3 mV after the loading of curcumin. This might be ascribed to the decrease of the negative charge density on FSG caused by the inner curled movement of gelatin. The *ζ*-potential of OFSG_2_ decreased from −12.2 to −21.0 mV after the loading of curcumin. This observation proved that more OFSGs might be attracted by curcumin, causing an increase of the negative charge density of the outer hydrophilic shell. Nevertheless, the significant change of the *ζ*-potential after the addition of curcumin might be ascribed to the rearrangement of gelatin molecules after curcumin attachment, which caused the exposure of negatively charged (OFSG) and positively charged (FSG) amino acids in the gelatin molecule [58]. The possible mechanism for the loading of curcumin into FSG or OFSG assembly is summarized in Figure 4.

### 3.4. In-vitro Simulated Gastrointestinal Stability and Release of Curcumin-Loaded FSG and OFSG

Fish (Giant catfish) skin gelatin was hydrolyzed by proteases, and an interesting observation was observed [59]. All hydrolysis processes were characterized by a high rate of hydrolysis (0% to 9%) during the initial stage (0–40 min), and then, a stationary stage was reached with further hydrolysis up to 90 min [59]. The rapid hydrolysis in the initial stage was owed to the large number of peptide bonds available [59], while the stationary hydrolysis in the second stage was mainly due to a decrease in the available substrate, enzyme autodigestion and product inhibition [60]. After the simulated gastro digestion, TEM observation showed that the procyanidin-gelatin nanoparticles lost their spherical shape to form irregular types and maintain their sizes of less than 100 nm without affecting the internal structure [61]. Silver carp scale gelatin also showed an excellent capacity of stabilizing oil-loaded emulsions during simulated gastrointestinal digestion [62]. However, whether the OFSG plays a protector of curcumin during the digestion tract still needs further verification. Thus, the retention and release of curcumin in the simulated gastrointestinal tract for curcumin-loaded FSG and OFSG were evaluated (Figure 5A–D).

The instability of curcumin during the variations of pH (The pH values of the SGF and SIF were 3.0 and 7.2) and the structural changes reflected in two aspects: (1) In an acidic pH environment, the free radical-driven incorporation of O_2_ that meets the criteria of an autoxidative process caused the autoxidation of curcumin into bicyclopentadione [40]. (2) In an alkaline pH environment, hydrolytic reactions happened that caused the degradation of curcumin into ferulic acid and feruloylmethane besides an oxidative reaction [63]. During the autoxidation of curcumin, H^+^ participated in the structure transition from a peroxyl radical to spiroepoxide. Thus, the permeability of H^+^ across the physical barrier constructed by FSG and OFSG determined the protection effect of self-assembly. Obviously, the loaded curcumin showed a faster degradation rate in the SIF than that in the SGF. At the end of the in vitro simulated gastric digestion, the retention of curcumin of the FSG and OFSG_1,2_ was 89%, 94% and 96%, respectively. At the end of the in vitro simulated intestinal digestion, the retention of curcumin of FSG and OFSG_1,2_ was 34%, 54% and 70%, respectively. The higher retention of curcumin in OFSG with a higher *DS* also proved that more OFSGs were surrounded the hydrophobic core. Finally, self-assembly constructed by OFSG exerted a better protective effect for curcumin both in the SGF and SIF. These phenomenon could be ascribed the following factors: Firstly, the acidic and alkaline pH environment contribute to the degradation of gelatin [64]. Thus, the hydrophobic area of FSG could easily be interrupted during the degradation of the molecular skeleton, while the hydrophobic cavity of OFSG was more stable with the aid of a hydrophobic interaction among the OSA groups. Secondly, the steric hindrance of the OSA in OFSG might prevent the enzyme actions of pepsin and pancreatin to hydrolyze gelatin molecules. Thirdly, the deprotonated carboxyl groups in the OFSG could neutralize the contacted H^+^ and stabilize the pH of the hydrophobic core [27]. Fourthly, the interaction with curcumin mostly occurred on the surface hydrophobic domains of the proteins [65]. Compared with the core–shell self-assembly, the outer location of gelatin for curcumin was more easily accessible to H^+^ or OH^-^. 

The release of curcumin for FSG and OFSG (1.1–2.2%) during 90 min of simulated gastric digestion was less than 3%, indicating that both the FSG and OFSG could successfully escort curcumin through the gastric tract, while the release of curcumin for FSG and OFSG during 360 min of simulated intestinal digestion showed two patterns. The release of curcumin for FSG gave a linear pattern, suggesting that the release of curcumin from these complexes was mainly dominated by the concentration-dependent diffusion from the polymer matrix [66]. The release of curcumin for OFSG_2_ firstly gave a linear pattern until 240 min and then kept steady. The first release pattern was due to the detachment of curcumin bounded to the surface hydrophobic area of gelatin, while the slower release after 240 min was mainly owed to the curcumin entrapped in the hydrophobic core of OFSG. Burst release [67] was not observed in release patterns, indicating a homogeneous distribution of curcumin in both the FSG and OFSG.

Three hypothetical curcumin release models were observed in related studies on the curcumin delivery system [63]: (1) burst releases, (2) prolonged release and (3) delayed release. The prolonged release model with outstanding bioavailability showed a moderate concentration of curcumin between the toxic and therapeutic levels in plasma. The OFSG showed a prolonged release model during the simulated intestinal digestion process, which was also observed in the release of curcumin by octenylsuccinate oat *β*-glucan [24] and (-)-epigallocatechin-3-gallate/poly(N-vinylpyrrolidone) nanoparticles [68].

## 4. Conclusions

The loading of curcumin by self-assembly constructed by OFSG was preliminary verified in this study. The most interesting point hinges on the association between CLE and the structural factor of the OFSG. The significant increase of CLE (μg/mL) from 6.98 to 24.67 μg/mL of the OFSG was mainly ascribed to two structural factors in two phases. The increase of CLE (μg/mL) of the OFSG was mainly due to the decrease of gelatin *M*_w_ during phase 1, depending on the degree of exposure of the interior hydrophobic groups in the advanced structure of FSG. The increase of CLE (μg/mL) of the OFSG was mainly due to the increase of *DS* during phase 2, depending on the increased molecular mobility and intermolecular hydrophobic interaction among OFSGs. FTIR and XRD of CL-OFSG showed a decreased absorption and crystallinity after the loading of curcumin. TEM gave an appearance of FSG in an aqueous system showing a large size (particle size, 1408 nm) and polygynous (PDI, 0.824) self-assembly, while the esterification of FSG led to the significant decrease of the particle size and PDI. After the loading of curcumin into OFSG assembly further facilitated the hydrophobic interaction, causing a significant decrease of the particle size and PDI and absorbing adjacent OFSGs to its surface (decrease of *ζ*-potential). The significantly higher retention of curcumin (54–70%) and lower release of curcumin (10.2–14.6%) at the end of the simulated intestinal digestion proved the self-assembly constructed by OFSG provided a more stable shell structure, protecting the degradation and preventing the escape of curcumin. The geneogenous deficiency of FSG, including the lower gelling point, melting point and gel strength, limits the possibilities for food development of the FSG compared with the gelatin from mammals. Therefore, the method of modification of the FSG taps a great potential for designing a carrier for delivering hydrophobic nutrients.

## Figures and Tables

**Figure 1 foods-11-02911-f001:**
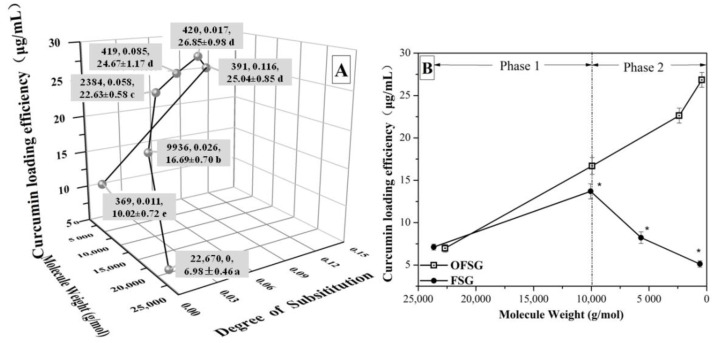
Curcumin loading efficiency (CLE, μg/mL) of fish scale gelatin (FSG) and octenylsuccinate fish scale gelatin (OFSG) with different molecular weights and degrees of substitution (**A**,**B**). CLE (**B**, μg/mL) bearing asterisks are significantly different (*p* < 0.05) between the OFSG and FSG with similar molecular weights.

**Figure 2 foods-11-02911-f002:**
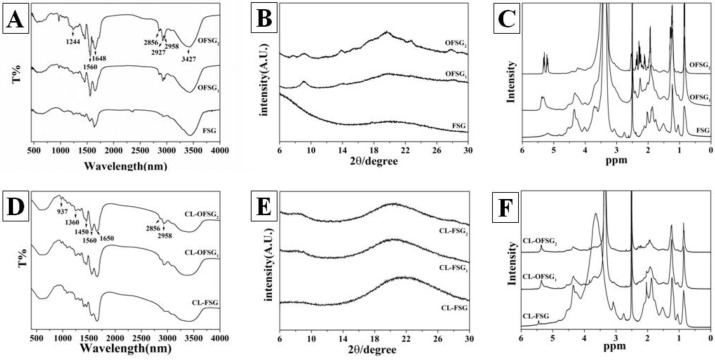
Fourier-transform infrared spectroscopy (FTIR, (**A**,**D**)), X-ray diffraction (XRD, (**B**,**E**)) and nuclear magnetic resonance (NMR, (**C**,**F**)) spectra of fish scale gelatin and octenylsuccinate fish scale gelatin (OFSG_1_, degree of substitution = 0.026; OFSG_2_, degree of substitution = 0.107) with or without the loading of curcumin.

**Figure 3 foods-11-02911-f003:**
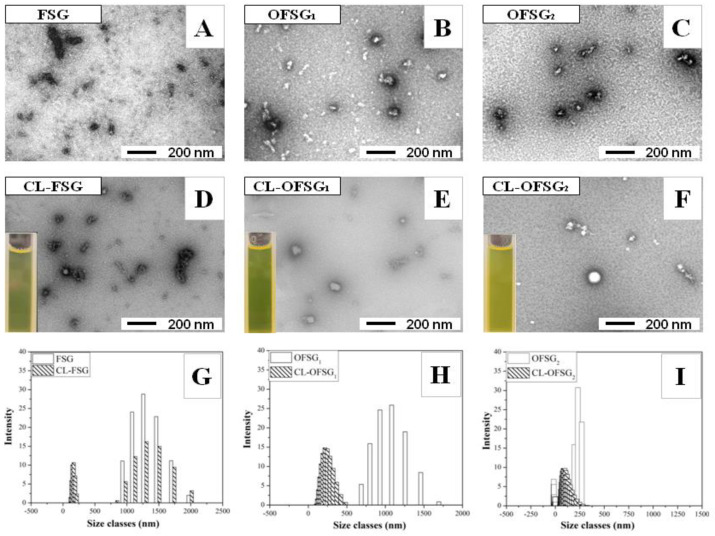
Transmission electron microscopy (TEM, **A**–**F**), particle size distribution (PSI, **G**–**I**) of fish scale gelatin (FSG) and octenylsuccinate fish scale gelatin (OFSG_1_, degree of substitution = 0.026; OFSG_2_, degree of substitution = 0.107) with (CL-FSG, CL-OFSG_1_ and CL-OFSG_2_) or without the loading of curcumin.

**Figure 4 foods-11-02911-f004:**
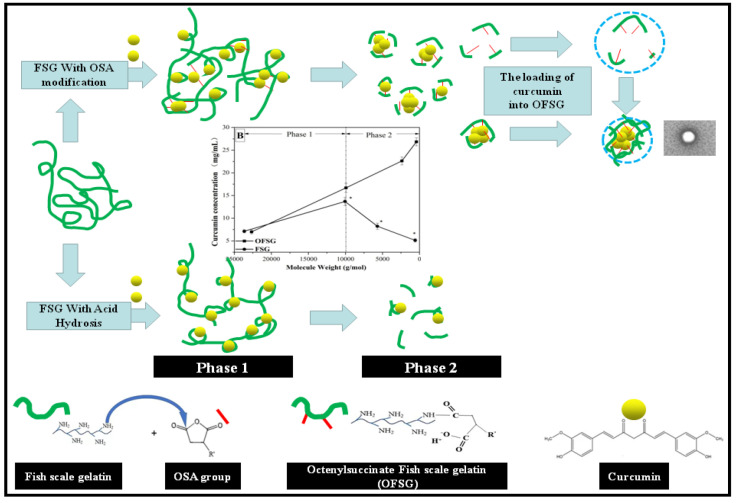
Schematic diagram illustrating the interactions between fish gelatin (FSG) with or without 2-octen-1-ylsuccinic anhydride (OSA) modification and curcumin.

**Figure 5 foods-11-02911-f005:**
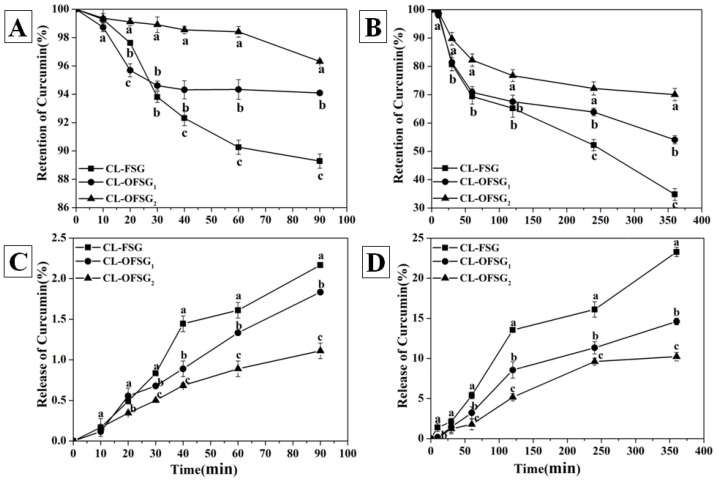
In vitro simulated gastric (**A**,**C**) and intestinal (**B**,**D**) stability and the release of curcumin-loaded fish scale gelatin (CL-FSG) and octenylsuccinate fish scale gelatin (CL-OFSG_1_, degree of substitution = 0.026; CL-OFSG_2_, degree of substitution = 0.107). Values bearing different superscript lowercase letters at the same time are significantly different (*p* < 0.05).

**Table 1 foods-11-02911-t001:** Percent contribution of the secondary structure of fish scale gelatin (FSG) and octenylsuccinate fish scale gelatin (OFSG). Values of samples bearing different superscript lowercase letters (a, ab, and b) within the same secondary structure are significantly different (*p* < 0.05).

Sample	Secondary Structure (%)			
	*α*-Helix	*β*-Sheet	*β*-Turn	Random Coil
FSG	13 ± 2 ^a^	52 ± 3 ^a^	20 ± 1 ^a^	15 ± 1 ^a^
OFSG_1_	27 ± 2 ^b^	35 ± 3 ^b^	22 ± 3 ^ab^	16 ± 2 ^a^
OFSG_2_	28 ± 1 ^b^	31 ± 4 ^b^	26 ± 2 ^b^	15 ± 1 ^a^

**Table 2 foods-11-02911-t002:** Particle size, particle distribution index (PDI) and *ζ*-potential of fish scale gelatin (FSG) and octenylsuccinate fish scale gelatin (OFSG_1_, degree of substitution = 0.026; OFSG_2_, degree of substitution = 0.107) with (CL-FSG, CL-OFSG_1_,and CL-OFSG_2_) or without the loading of curcumin. Values of samples bearing different superscript lowercase letters (a, b, c, d, and e) within the particle size, PDI and *ζ*-potential are significantly different (*p* < 0.05).

Sample	Particle Size (nm)	PDI	*ζ*-Potential (mV)
FSG	1408.3 ± 84.0 ^a^	0.824 ± 0.070 ^a^	−25.2 ± 2.27 ^c^
OFSG_1_	1270.7 ± 79.5 ^b^	0.659 ± 0.063 ^b^	−11.4 ± 0.73 ^a^
OFSG_2_	350.0 ± 31.7 ^d^	0.584 ± 0.048 ^b^	−12.2 ± 1.31 ^a^
CL-FSG	575.5 ± 65.4 ^c^	0.427 ± 0.033 ^c^	−18.3 ± 0.97 ^b^
CL-OFSG_1_	214.5 ± 32.3 ^e^	0.347 ± 0.012 ^c^	−20.3 ± 2.02 ^b^
CL-OFSG_2_	139.5 ± 30.6 ^e^	0.248 ± 0.022 ^d^	−21.0 ± 2.04 ^b^

## Data Availability

The data presented in this study are available on request from the corresponding author.

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
