# Peer review of "Curcumin-Loaded Self-Assembly Constructed by Octenylsuccinate Fish (Cyprinus carpio L.) Scale Gelatin: Preparation and Characterization"

_foods, 2022, doi:10.3390/foods11182911_

Round 1

Reviewer 1 Report

Authors should describe in detail the effect of  octenyl succinate on the shape of particles and variations in particle size compared to FSG. Also, inclusion of chemical structure of model drug and modification moiety should be added.  

Author Response

Reviewer 1:

1  Authors should describe in detail the effect of octenyl succinate on the shape of particles and variations in particle size compared to FSG. Also, inclusion of chemical structure of model drug and modification moiety should be added. 

Response: Thanks a lot for the helpful suggestion. We had rewrote the description of the effect of octenyl succinate on the shape of particles and variations in particle size compared to FSG as “The self-aggregation of FSG and OFSG with or without the loading of curcumin was further characterized using TEM and dynamic light scattering (DLS) technique (Fig.3 and Tab. 1). The appearance of FSG self-assembly (Fig.3 A) was obviously different from OFSG1-2 self-assembles (Fig.3 B-C). Self-assembly formed with FSG showed a solid structural particle with non-uniform distribution and without hollow core. While the grafting of OSA into FSG caused the formation of nanoparticles with core-shell structure (Fig.3 B and C) and a narrower particle size distribution (PSD) from 1000-2000 nm to 125-275 nm (Fig.3 G-I). Particle size, particle size distribution (PDI) and ζ-potential of FSG (Table 1) were 1408.3 nm, 0.824 and -25.2 mV, respectively, which was in accordance with the appearance of TEM result showing a large size and polygonous self-assembly. While the grafting of OSA groups into FSG exerted a significant effect on the particle size, PDI and ζ-potential of aggregate. The increase of DS from 0 to 0.107 caused the significant decrease of particle size from 1408 nm to 350 nm and PDI from 0.824 to 0.584. This variation of self-aggregate shape and homogeneity was ascribed to the grafting of OSA groups into FSG, which caused the enhancement of hydrophobic interaction among OFSGs [57]. While the decrease of Mw during the chemical modification also contributed to the decrease of particle size and PDI. Since polymer with lower Mw possessed lower steric hindrance and higher flexibility, facilitating access to self-assemblies [27]. The ζ-potential, which provided the information on the surface properties of self-aggregates, increased from -25.2 to -12.2 with the increase of DS from 0 to 0.107. The decrease of ζ-potential probably ascribed to 2 reason: 1) the significant decrease of Mw from 22677 to 420 g/mol, which accompanied with the decrease of negative charge density on the surface of self-aggregates. 2) the formation of new NH3+ groups as a result of hydrolysis of gelatin molecules. The TEM of OFSG also presented self-aggregates with more homogeneity after the grafting of OSA groups”. (Please see Line 399-424)

Nevertheless, the chemical structure of model drug and modification moiety had been supplemented in the Fig.4. (Please see Figure4 )

Reviewer 2 Report

The manuscript “Curcumin-loaded self-assembly Constructed by Octenylsuccinate Fish (Cyprinus carpio L.) Scale Gelatin: Preparation and Characterization” describes the preparation of curcumin loaded octenylsuccinate fish scale gelatin and characterizations to evaluate the feasibility for applying as a carrier for hydrophobic nutrients in food and pharmaceutical industries. The authors prepared various types of fish scale gelatin based on adding curcumin and octenylsuccinate and found some interesting results. A variety of physico-chemical characteristics of the prepared samples such as degree of substitution (DS), curcumin loading efficiency, FTIR, XRD, PDI, particle zixe, ζ-potential etc. were evaluated. The manuscript is well written, well-balanced, and nicely described. The scientific value and language is sound. The approach and study used in this study is novel and the contents of this paper fit the scope of this journal. However, there are some minor issues to follow corrections.

Specific Comments:

1.      Abstract: Please write a statement mentioning the aim of the study.

2.      Abstract: Line 21; replace the word “caused” by “showed”

3.      Introduction: Line 31; “The biomaterial in FS is mainly composed of type â…  collagen, which endows its application potential in food, medicine, chemical engineering, and other industries”. Please add reference supporting the statement.

4.      Materials and Methods: Line 106; “The synthesis of FSG was conducted as the method described before...” should be “The synthesis of octenylsuccinate based FSG was conducted as the method described before...”

5.      Figure 1-4: Please enlarge the size of Figures. Too small for the readers to visualize. I suggest showing the tables separately from Fig. 1 and Fig. 3.

Author Response

Reviewer 2:

1  Abstract: Please write a statement mentioning the aim of the study.

Response: Thanks a lot for the helpful suggestion and the statement had been supplemented in the manuscript as “Curcumin loaded octenylsuccinate fish scale gelatin (OFSG) was prepared in this study, to explore the potential of FSG for delivering hydrophobic nutrient”. (Please see Line 16-17)

2  Abstract: Line 21; replace the word “caused” by “showed”

Response: We apologize for the wrong expression and the ward “caused” had been corrected as “showed”. (Please see Line 21)

3  Introduction: Line 31; “The biomaterial in FS is mainly composed of type â…  collagen, which endows its application potential in food, medicine, chemical engineering, and other industries”. Please add reference supporting the statement.

Response: We apologize for the missing of reference for the statement and the related ref. had been supplemented in the manuscript. (Please see Line 31-33

4  Materials and Methods: Line 106; “The synthesis of FSG was conducted as the method described before...” should be “The synthesis of octenylsuccinate based FSG was conducted as the method described before...”

Response: We apologize for the wrong expression and the sentence had been corrected. (Please see Line 111-112)

5  Figure 1-4: Please enlarge the size of Figures. Too small for the readers to visualize. I suggest showing the tables separately from Fig. 1 and Fig. 3.

Response: We apologize for the inadequate arrangement of Figures and Tables. They had been arranged as the suggestion provided by the reviewer. (Please see Figure 1-4 and Table 1-2)

Reviewer 3 Report

Review foods 1851608

The characteristic of gelatin from fishery sources should be discussed further. The contribution from their specific structure and properties should be emphasized. For instance, Food Hydrocolloids, 94, 459-467; Food Hydrocolloids, 90, 9-18; Journal of Food Engineering, 239, 92-103.  ​

The characteristic wavenumber which should be linked to the specific structure and properties of the fish gelatin should be discussed further. For instance, Food Hydrocolloids, 45, 72-82.   

Figure 4: after digestion, why did the fish gelatin still there? Further elaboration is needed. 

Figure 2: for the standard at 0 ppm, what was applied? TSP or others? Which information should be critical for better understanding the results? For instance, Food Chemistry, 286, 87-97. 

For the release system, what was the fundamental difference from previous report? For instance, Biomaterials Advances, 134, 112717. 

Author Response

Reviewer 3:

1  The characteristic of gelatin from fishery sources should be discussed further. The contribution from their specific structure and properties should be emphasized. For instance, Food Hydrocolloids, 94, 459-467; Food Hydrocolloids, 90, 9-18; Journal of Food Engineering, 239, 92-103.  â€‹

Response: Thanks a lot for the helpful suggestion, and the discussion had been supplemented in the introduction as “Normally, the application of gel formed with fish gelatin (FG) was limited due to their lower strength, gelling and melting temperature. Thus, physicochemical properties of FG from skin or scale could be enhanced by modification with acyl gellan [6], sodium alginate [7]and κ-carrageenan [8].” (Please see Line 42-45)

Newly added Ref.

[6]  Sow, L. C., Tan, S. J., & Yang, H. Rheological properties and structure modification in liquid and gel of tilapia skin gelatin by the addition of low acyl gellan. Food Hydrocolloids 2019, 90, 9-18.

[7]  Sow, L. C., Toh, N. Z. Y., Wong, C. W., & Yang, H. Combination of sodium alginate with tilapia fish gelatin for improved texture properties and nanostructure modification. Food Hydrocolloids 2019, 94, 459-467.

[8]  Sow, L. C., Chong, J. M. N., Liao, Q. X., & Yang, H. Effects of κ-carrageenan on the structure and rheological properties of fish gelatin. Journal of Food Engineering 2018, 239, 92-103.

2 The characteristic wavenumber which should be linked to the specific structure and properties of the fish gelatin should be discussed further. For instance, Food Hydrocolloids, 45, 72-82.   

Response: Thanks a lot for the helpful suggestion, and the discussion the specific structure and properties of the fish gelatin had been supplemented in the manuscript as “FTIR spectra of FSG showed characteristic amide-I, amide-â…¡and amide-III bands at approximately 1633, 1560 and 1244 cm−1 [45]. The location of amide-I was shifted from 1633 cm−1 to 1648 cm−1, after OSA modification of FSG. This variation might indicate the change in the secondary structure of gelatin [46]. To further comprehend the change of gelatin structure after the OSA modification, curve fitting was semi-empirically processed based on the reported results (Supplementary 1 A-C). Several shoulder peaks indicating the secondary structures of protein hid in amide I band, and could be estimated by the aid of deconvolution of amide I [47]. Secondary structure including α-helix, β-sheet, β-turn, and random coil were assigned to band area I (between 1654 and 1658 cm-1), band area â…¡( between 1642, 1638, 1632, 1627 and 1624 cm-1) , band location I (at 1,688, 1,680, 1,672 and 1,666 cm-1), and band location â…¡(at 1,648 cm-1), respectively [48]. The results after calculation had been recorded in Tab. 1. There are 4 kinds of basic secondary structure (α-helix, β-sheet, β-turn, and random coil) existed in FSG and OFSG. Among which, random coil did not show significant difference between FSG and OFSG. While, the significant increase of α-helix from 13% to 28%, decrease of β-sheet from 52% to 31% and increase of β-turn from 20% to 26% had been observed (Tab.1). However, grafting of OSA to FSG molecule and hydrolyzing of FSG took place simultaneously, during the chemical modification. Thus, the change of secondary structure of FSG was caused by a complex reaction system (both decrease of Mw and increase of DS during the modification). The transition among 4 different secondary structure of peptide could affect the final formation of nanostructures through diverse self-assembly pathways [49]. Thus, the transition of β-sheet to α-helix might also contribute to the self-assembly of OFSG”. (Please see Line 316-399)

Newly added Ref.

[45]       Uriarte-Montoya, M. H., Santacruz-Ortega, H., Cinco-Moroyoqui, F. J., RouzaudSández, O., Plascencia-Jatomea, M., & Ezquerra-Brauer, J. M. Giant squid skin gelatin: chemical composition and biophysical characterization. Food Research International 2011, 44(10), 3243-3249.

[46]       Sow, L. C., & Yang, H. Effects of salt and sugar addition on the physicochemical properties and nanostructure of fish gelatin. Food Hydrocolloids 2015, 45, 72-82.

[47]       Byler, D. M., & Susi, H. Examination of the secondary structure of proteins by deconvolved FTIR spectra. Biopolymers 1986,25, 469-487

[48]       Yang, H., Yang, S., Kong, J., Dong, A., & Yu, S. (2015). Obtaining information about protein secondary structures in aqueous solution using Fourier transform IR spectroscopy. Nature protocols 2015,10(3), 382-396.

[49]       Ghosh, G., Barman, R., Mukherjee, A., Ghosh, U., Ghosh, S., & Fernandez, G. Control over Multiple Nano-and Secondary Structures in Peptide Self-Assembly. Angewandte Chemie 2022,1(5), e202113403.

3 Figure 4: after digestion, why did the fish gelatin still there? Further elaboration is needed. 

Response: Thanks a lot for reviewer’s attention on the anti-digestion of FSG or OFSG. After careful checking up in the former study, we had concluded the related elaboration in the manuscript as “Fish (Giant catfish) skin gelatin was hydrolyzed by proteases and an interesting observation was observed [60]. All hydrolysis processes were characterised by a high rate of hydrolysis (0% to 9%) during the initial stage (0–40 min), and then a stationary stage was reached with further hydrolysis to 90 min [60]. The rapid hydrolysis in the initial stage was owing to a large number of peptide bonds available [60]. While, the stationary hydrolysis in the second stage was mainly due to a decrease in available substrate, enzyme auto-digestion and product inhibition [61]. After the simulated gastro digestion, TEM observation showed that procyanidins-gelatin nanoparticles lost their spherical shape to forming irregular type and maintaining their sizes of less than 100 nm without affecting the internal structure [62]. Silver carp scale gelatin was also showed an excellent capacity of stabilizing oil-loaded emulsion during in simulated gastrointestinal digestion [63]. However, whether OFSG plays a protector of curcumin during the digestion tract still need further verification. Thus, the retention and release of curcumin in simulated gastrointestinal tract for curcumin loaded FSG and OFSG was evaluated (Fig.5 A-D)”. (Please see Line 472-486)

Newly added Ref.

[60]       Ketnawa, S., Martínez-Alvarez, O., Benjakul, S., & Rawdkuen, S. Gelatin hydrolysates from farmed Giant catfish skin using alkaline proteases and its antioxidative function of simulated gastro-intestinal digestion. Food Chemistry 2016, 34-42.

[61]       Khantaphant, S., Benjakula, S., & Kishimura, H. Antioxidative and ACE inhibitory activities of protein hydrolysates from the muscle of brownstripe red snapper prepared using pyloric caeca and commercial proteases. Process Biochemistry 2011, 318–327.

[62]       Carmelo-Luna, F. J., Mendoza-Wilson, A. M., Montfort, G. R. C., Lizardi-Mendoza, J., Madera-Santana, T., Lardizábal-Gutiérrez, D., & Quintana-Owen, P. Synthesis and experimental/computational characterization of sorghum procyanidins-gelatin nanoparticles. Bioorganic & Medicinal Chemistry 2021, 116240.

[63]       Xu, J., Huang, S., Zhang, Y., Zheng, Y., Shi, W., Wang, X., & Zhong, J. Effects of Antioxidant Types on the Stabilization and In Vitro Digestion Behaviors of Silver Carp Scale Gelatin-Stabilized Fish Oil-Loaded Emulsions. Colloids and Surfaces B: Biointerfaces 2022, 112624.

4 Figure 2: for the standard at 0 ppm, what was applied? TSP or others? Which information should be critical for better understanding the results? For instance, Food Chemistry, 286, 87-97. 

Response:  We apologize for the missing information of the used standard in NMR and it had been supplemented in the manuscript as “tetramethylsilane (TMS) was used as standard at 0 ppm”. (Please see Line 191)

Newly added Ref.

[30]       Chen, L., Wu, J. E., Li, Z., Liu, Q., Zhao, X., & Yang, H. Metabolomic analysis of energy regulated germination and sprouting of organic mung bean (Vigna radiata) using NMR spectroscopy. Food Chemistry 2019, 286, 87-97.

5 For the release system, what was the fundamental difference from previous report? For instance, Biomaterials Advances, 134, 112717. 

Response: The release system in the previous study (Biomaterials Advances, 134, 112717) was based on the insolubilized solid gel where curcumin molecule was absorbed or trapped in the hole on the surface of mixed gel causing the inhomogeneous distribution of curcumin. Nevertheless, there wasn’t any obstacle between the loaded curcumin and the receiving media. Thus, a burst release of curcumin was observed in the final test. While, the release system in our study was based on the solubilized gel molecules where curcumin molecule was trapped in the hollow cavity of OFSA causing the homogeneous distribution of curcumin. Thus, a sustain release of curcumin was observed in the final test.

Round 2

Reviewer 3 Report

Review foods-1851608

The authors have addressed the questions quite well. The revised manuscript has been improved significantly. There are no further comments. The current version is acceptable for publication.

Author Response

Thanks.